# Assessment of Aging Impact on Wax Crystallization in Selected Asphalt Binders

**DOI:** 10.3390/ma15228248

**Published:** 2022-11-21

**Authors:** Wenqi Wang, Ali Rahman, Haibo Ding, Yanjun Qiu

**Affiliations:** 1School of Architecture and Civil Engineering, Xihua University, Chengdu 610039, China; 2School of Civil Engineering, Southwest Jiaotong University, Chengdu 610031, China; 3Highway Engineering Key Laboratory of Sichuan Province, Southwest Jiaotong University, Chengdu 610031, China

**Keywords:** asphalt binder, aging, oxidation, wax, nuclear magnetic resonance, gas chromatography

## Abstract

For a better understanding of the changing trend in crystalline components of asphalt binders, asphalt binders originating from the SHRP Materials Reference Library with different oxidation degrees (unaged, 20 h PAV, and 60 h PAV) were prepared. The native asphalt binders and their oxidized residues were characterized by liquid-state nuclear magnetic resonance (NMR) spectroscopy and high-temperature gas chromatography (HTGC). The results showed that, compared with other carbon types, the content of internal methylene carbons of long paraffinic chains between different SHRP binders was quite different. The NMR average length of a long paraffinic internal methylene chain showed a good correlation with the wax content obtained at −20 °C using the methyl ethyl ketone (MEK) precipitation method and also the recently developed variable-temperature Fourier-transform infrared spectroscopy (VT-FTIR) method. In most cases, the average length of straight internal methylene carbons of a long paraffinic chain terminated by a methyl group increased with the oxidation of the asphalt binder. However, the difference caused by oxidation was significantly smaller than the difference caused by the source of the asphalt binder. In general, oxidation will make the n-alkanes distributed in asphalt binder fall within a narrower range. The carbon number of n-alkanes in the asphalt binder generally grew with oxidation.

## 1. Introduction

The quality of asphalt binder is one of the important factors affecting the durability of asphalt pavement. In the past, researchers tried to establish a direct link between the chemical composition of asphalt and its quality; however, they failed to do so. This was mainly due to the complexity of asphalt chemical composition and a lack of in-depth understanding of the asphalt chemical composition’s classification. Among the chemical components of asphalt, wax is undoubtedly an important substance affecting the low-temperature service performance of asphalt [1,2,3,4]. Consequently, wax content and its behavior in asphalt with thermal history have received considerable attention [5,6,7]. Wax is generally considered to be a substance that crystallizes in asphalt at 25 °C and is mainly composed of saturated long linear hydrocarbons [8]. Owing to wax self-polymorphism and the difference in the state of existence in asphalt, there is no consensus on the wax effect on asphalt performance. This is one of the reasons why countries imposed different limits on wax content in asphalt.

Separating wax is the most direct method to study wax structure in asphalt [9]. Several methods were proposed to physically separate the wax from the asphalt binder. Due to the interference of the polar components (such as asphaltenes and resins) in the asphalt, it is difficult to separate the wax directly from the asphalt binder. Therefore, distillation or solvent precipitation is usually utilized to remove asphaltenes from the asphalt binder. The remaining components (maltenes) continue to be separated using chromatographic techniques. According to Corbett and Swarbrik’s method [10], the maltenes fraction is adsorbed on a chromatographic column (alumina is used as the adsorbent phase) and sequentially desorbed with solvents of increasing polarity. Saturates, aromatics, and resins are obtained from the maltenes. Using methyl ethyl ketone-benzene as a dewaxing solvent, solid wax can be separated from saturated and aromatic fractions at low temperatures. In contrast, Rostler and White [11] separated the maltenes into nitrogen bases, first acidaffins, second acidaffins, and paraffin fractions based on their reactivity with sulfuric acid (decreasing degree of hydration). The last fraction is what is called “wax”. In some studies, asphalt’s neutral fraction obtained from ion exchange chromatography (IEC) is regarded as a “wax fraction” [12]. When preparative size exclusion chromatography (SEC) is utilized to separate asphalt binder, the wax component is in the SEC-II fractions (non-associating components) [13]. Waxes are mainly composed of normal alkanes and isomeric alkanes, and the former produces the most adverse effect on asphalt performance. Considering urea readily forms crystalline adducts with straight-chain hydrocarbons, Netzel et al. [14] applied this technique to determine the content of n-alkanes and regarded this as solid wax content in asphalt.

With the development of modern analytical instruments, various advanced physical and chemical characterization methods have been employed to study wax behavior in asphalt [15,16,17,18]. Differential scanning calorimetry (DSC) is a common analytical method to study the crystalline fractions in asphalt binder. The number of crystallizable fractions in asphalt binder can be quantified by the size of the endothermic peak during heating. Using a thermal analysis of asphalt components, Corbett et al. [19] concluded that the endothermic peak of asphalt was mainly caused by saturates. However, Harrison et al. [20] believed that the saturates fraction was not the main factor leading to the endothermic behavior of asphalt binder, and the average linear side chain length was the main factor that affected the size of the endothermic peak. The effects of oxidation on the thermal behavior of pure wax-doped asphalt systems were studied by Kovinich et al. [6]. They found that the size of the endothermic peaks in the heat flow curve decreased with the oxidation of the binder. Atomic force microscopy (AFM) is often applied to observe the morphology of asphalt and the “bee structures” found in asphalt are thought to be the wax. Lu et al. [21] showed that the bee structures were fewer but larger after prolonged oxidation. They attributed this phenomenon to the highly increased stiffness of aged binder and reduced compatibility between the crystalline fraction and the more polarized asphalt matrix. Recently, Ding et al. [5,22,23] developed a variable-temperature Fourier-transform infrared spectroscopy technique to study the wax in asphalt, including quantification of the wax content and the precipitation and melting process of wax in asphalt. Aging is an important factor leading to the deterioration of asphalt performance. There have been a lot of literature reports on the mechanism and chemical reaction path of asphalt aging [24,25,26,27,28]. However, they did not find a clear rule regarding the effects of oxidation on wax content in asphalt binder.

As can be seen from the above literature review, although the wax in asphalt has been studied to a certain degree, the effect mechanisms of oxidative aging on wax in asphalt are still unclear. In this study, various asphalt binders from different crude oil sources were oxidized to a different degree in the laboratory. High-temperature gas chromatography (HTGC) and liquid-state nuclear magnetic resonance (NMR) spectroscopy were employed to analyze the structure of asphalt binders under different aging conditions. It is expected that some new insights into oxidation’s effects on the crystallizable fractions in asphalt could be provided by a series of laboratory tests.

## 2. Materials and Methods

### 2.1. Asphalt Binders

In this study, eight asphalt binders from the SHRP (Strategic Highway Research Program) materials reference library were utilized. The SHRP materials reference library was built during the SHRP and used to store the asphalt binders adopted in the implementation of the project. The physical and chemical properties of these asphalt binders have been extensively studied to reveal the mechanisms of differences in field performance. Due to the fact that the SHRP binders have different colloidal structures and crude oil sources, they are representative and generally applied as base binders for modified asphalts to validate the performance of the modifier. The short-term aging of asphalt was simulated by Rolling Thin-Film Oven (RTFO) experiments, and the long-term aging behavior of asphalt was simulated by a Pressure Aging Vessel (PAV). In addition to the traditional 20-h PAV test, 60 h oxidative aging time was also carried out to better simulate the degree of field asphalt. In a previous study [5], although the effects of oxidation on the wax contents of these SHRP binders were studied using the VT-FTIR method, no precise rule was found. This study adopts the same SHRP binders and aging modes but with different characterization methods. In this way, the obtained results could be mutually confirmed and compared. The basic properties of the selected SHRP binders can be found in the reference [23,29].

### 2.2. Nuclear Magnetic Resonance (NMR)

Nuclear magnetic resonance spectroscopy can provide information about the relative number of different types of hydrogen, carbon, or other atoms in a molecule, and the chemical environment in which these different types of atoms are exposed. The abscissa of the NMR spectrum is the chemical shift, and the ordinate is the intensity of the resonance absorption peak, whose peak area is proportional to the number of such nuclei in the molecule. Since the absolute value of the chemical shift of the general atomic nucleus is very small, the peak of tetramethylsilane (TMS) is usually utilized as a reference for comparison, and the relative change value is expressed as ppm (parts per million). In the case of mixtures such as asphalt, NMR provides average structure information for the entire sample. This information is useful in characterizing and differentiating the chemical properties of different asphalts as a whole. The nuclear overhauser effect (NOE) occurs due to the influence of ^1^H adjacent to ^13^C, which can significantly enhance the sensitivity of ^13^C NMR. However, its influence on different types of carbon is inconsistent and could result in the spectrum’s intensity not being proportional to the carbon amount. For this reason, this study adopted the measures of inverse gated decoupling and relaxation reagent to eliminate the NOE. In this way, a quantitative determination based on the integrated intensity of the spectrum could be carried out.

### 2.3. Gas Chromatography

Gas chromatography is used to separate the components of a sample with different partitioning coefficients between the mobile phase and stationary phase of the column [30]. In other words, the mobile phase has different adsorption values or solubility over the stationary phase. The mixture about to be separated is vaporized into a gas at the injection port, and the gas is carried by the mobile phase into the chromatographic column. Owing to the continuous flushing of the mobile phase, the components move downstream, and the components with the weakest adsorption (or dissolution) ability move downstream faster than other components. In this way, each component in the sample could flow out from respective weak to strong adsorption according to its adsorption (or dissolution) ability, so that the components can be separated in this way. The detector response versus retention time is the raw data obtained from the gas chromatography test. The time from injection to the chromatographic peak reaching the local maximum value is called the retention time of the component. Under a specific stationary phase and experimental conditions, each component has a specific retention time, which could be used as a qualitative indicator. The peak height reflects the response of the detector.

## 3. Results and Discussion

### 3.1. ^13^C NMR Spectra of Asphalt Binders

The ^13^C NMR spectra of SHRP asphalt binders under different oxidation states (unaged, 20 h PAV, and 60 h PAV) are presented in Figure 1. According to previous studies, the peaks between 110–160 ppm were attributed to aromatic carbons, and the peaks between 10–40 ppm were attributed to carbons in aliphatic chains and naphthenic rings [31]. Considering the objectives of this study, only aliphatic carbons were analyzed. It can be seen from the ^13^C NMR spectrum of unaged SHRP asphalt binders that all tested asphalts have five obvious main peaks. It was reported that the peak at 14.1 ppm was assigned to methyl carbons of aliphatic side chains [32]. The peaks at 22.7, 31.9, and 29.1 ppm correspond to carbons in methylene units successively further from the methyl group. The peak at 29.7 ppm, which is the largest peak in the aliphatic carbon region, is caused by internal methylene carbons of long paraffinic chains.

It is difficult to distinguish the difference between ^13^C spectra of various SHRP asphalt binders with the naked eye. In addition, the influence of the degree of oxidative aging in SHRP binders on the ^13^C chemical shift is also difficult to be evaluated directly. Therefore, the ratio of methylene in different positions to the 14.1 ppm methyl peak is presented in Figure 2. The ratio is an indication of the average length of a straight methylene chain terminated by a methyl group [33]. Compared with internal methylene carbons of long paraffinic chains, oxidative aging does not produce a major effect on the contents of methylene carbons adjacent to the methyl functional group in the average molecular structure of the asphalt binder. Moreover, in comparison with other carbon types, the content of internal methylene carbons of long paraffinic chains between various SHRP binders is quite different.

In order to determine whether the ^13^C spectra signal can be utilized to determine the wax content in the asphalt binder, the NMR average length of a long paraffinic internal methylene chain was compared with the wax content obtained at −20 °C using the methyl ethyl ketone (MEK) precipitation method and also the recently developed variable-temperature Fourier-transform infrared spectroscopy (VT-FTIR) method, respectively. Corresponding results are illustrated in Figure 3. It can be seen that the NMR signal shows a close correlation with the other two selected methods, especially the VT-FTIR method. Compared with the commonly used chemical precipitation method to determine the wax content, the NMR method has many merits, including straightforward steps, less effect on results by the operating factors, and usage of almost no toxic chemical solvents. The most important fact is that this method can effectively distinguish asphalt binders with different wax contents from each other. However, compared with VT-FTIR, the results obtained by the NMR method are less discriminating. The difference between the highest and lowest wax content obtained using the VT-FTIR method is up to 9.07, while it is only 4.5 using the NMR method. This may be due to the fact that NMR is tested at room temperature and deuterated chloroform is used as the solvent. The presence of solvents may destroy the form of the molecular structure in the asphalt matrix. In the future, low-temperature solid-state nuclear magnetic resonance spectroscopy will be employed to further test asphalt binders and verify the accuracy of the conjecture. Similar to a previous publication by Ding et al. [5], the NMR method has not found a clear rule of the effect of oxidation on the crystallizable fractions in asphalt binder. In most cases, the average length of straight internal methylene carbons of a long paraffinic chain terminated by a methyl group increases with the oxidation of the asphalt binder. However, the difference caused by oxidation is much smaller than the source of asphalt binder.

### 3.2. DEPT 135 Spectra of Asphalt Binders

The main function of using the distortionless enhancement by polarization transfer (DEPT) test is to distinguish methyl (CH_3_), methylene (CH_2_), and methine (CH) groups and clarify the aliphatic region of ^13^C spectra. In general, methyl and methine carbons are displayed as upward signals in the DEPT 135 experiment, whereas the methylene carbons appear as downward signals. DEPT 135 spectra of sample AAV and AAC-1 under three oxidation states (unaged, 20 h PAV, and 60 h PAV) are shown in Figure 4. As expected, the peaks at 22.7, 31.9, 29.1, and 29.7 ppm correspond to carbons in methylene units successively further from the methyl group, and internal methylene carbons of long paraffinic chains are downward. The peaks at 14.1 ppm and 19.71 ppm correspond to methyl peaks, and methyl branches on a straight methylene backbone are upward. The aromatic/aliphatic carbons ratio in asphalt binder can also be determined using the DEPT 135 NMR experiment. Considering the main purpose of this study, the corresponding results are not shown here.

### 3.3. Effect of Oxidation on n-Alkane Distribution

Under the selected optimal chromatographic analysis conditions, the carbon disulfide solutions of asphalt with different aging states (unaged, 20 h PAV, and 60 h PAV) were analyzed and determined. The results of removing the solvent peaks are presented in Figure 5. Since the correction factors of hydrocarbon compounds on the flame ionization detector (FID) are similar, the area normalization method is directly used for quantitative analysis to obtain the content of each n-alkane. The determination results of carbon number distribution are illustrated in Figure 6. Although the effects of oxidation on the total n-alkane content in asphalt cannot be determined by gas chromatogram, some interesting observations could be made. For the unaged ABG asphalt sample, the carbon number of n-alkanes is mainly distributed in a wide range between 30 and 80. Moreover, there are very few n-alkanes with carbon numbers below 20 and above 90. With the oxidation of the binder (20 h PAV), the carbon number of n-alkanes in the asphalt is mainly concentrated in a narrow range between 35 and 50. The sample hardly contains n-alkanes with carbon numbers lower than 30 and higher than 80. By extending the binder’s oxidation time (60 h PAV), the carbon number of n-alkanes in asphalt generally increased. A similar trend was found in other asphalt binders. Furthermore, the carbon number distribution of n-alkanes tends to be uniform, especially for AAM-1 and AAC-1.

## 4. Summary and Conclusions

In this study, the effects of oxidation on crystallizable fractions in different asphalt binders were investigated by liquid-state nuclear magnetic resonance (NMR) spectroscopy and high-temperature gas chromatography (HTGC). The following conclusions can be drawn from a series of laboratory tests:(1)The results of the ^13^C NMR spectra of unaged SHRP asphalt binders revealed that all tested asphalts had five obvious main peaks. Compared with internal methylene carbons of long paraffinic chains, oxidative aging did not produce a major effect on the contents of methylene carbons adjacent to the methyl functional group in the average molecular structure of the asphalt binder.(2)Compared with other carbon types, the content of internal methylene carbons of long paraffinic chains between different SHRP binders was quite different. The NMR average length of a long paraffinic internal methylene chain showed a close correlation with wax content obtained at −20 °C using the methyl ethyl ketone (MEK) precipitation method and also the recently developed variable-temperature Fourier-transform infrared spectroscopy (VT-FTIR) method.(3)The application of the NMR method could not find a clear rule in terms of the effect of oxidation on the crystallizable fractions in asphalt binder. In most cases, the average length of straight internal methylene carbons of a long paraffinic chain terminated by a methyl group increased with the oxidation of the asphalt binder. However, the difference caused by oxidation was considerably smaller than the source of asphalt binder.(4)In general, oxidation will cause the n-alkanes distributed in asphalt binder to fall within a narrower range. The carbon number of n-alkanes in asphalt binder generally increases with oxidation.(5)Future studies will cover the evaluation of UV effects, the short-term RTFOT test, as well as the determination of rheological properties.

## Figures and Tables

**Figure 1 materials-15-08248-f001:**
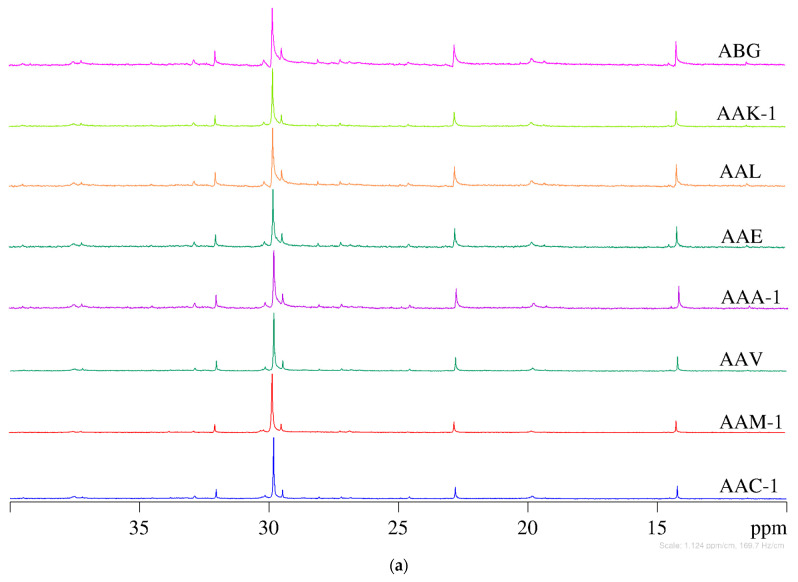
^13^C NMR spectra of SHRP asphalt binders. (**a**) Unaged, (**b**) 20 h PAV, and (**c**) 60 h PAV.

**Figure 2 materials-15-08248-f002:**
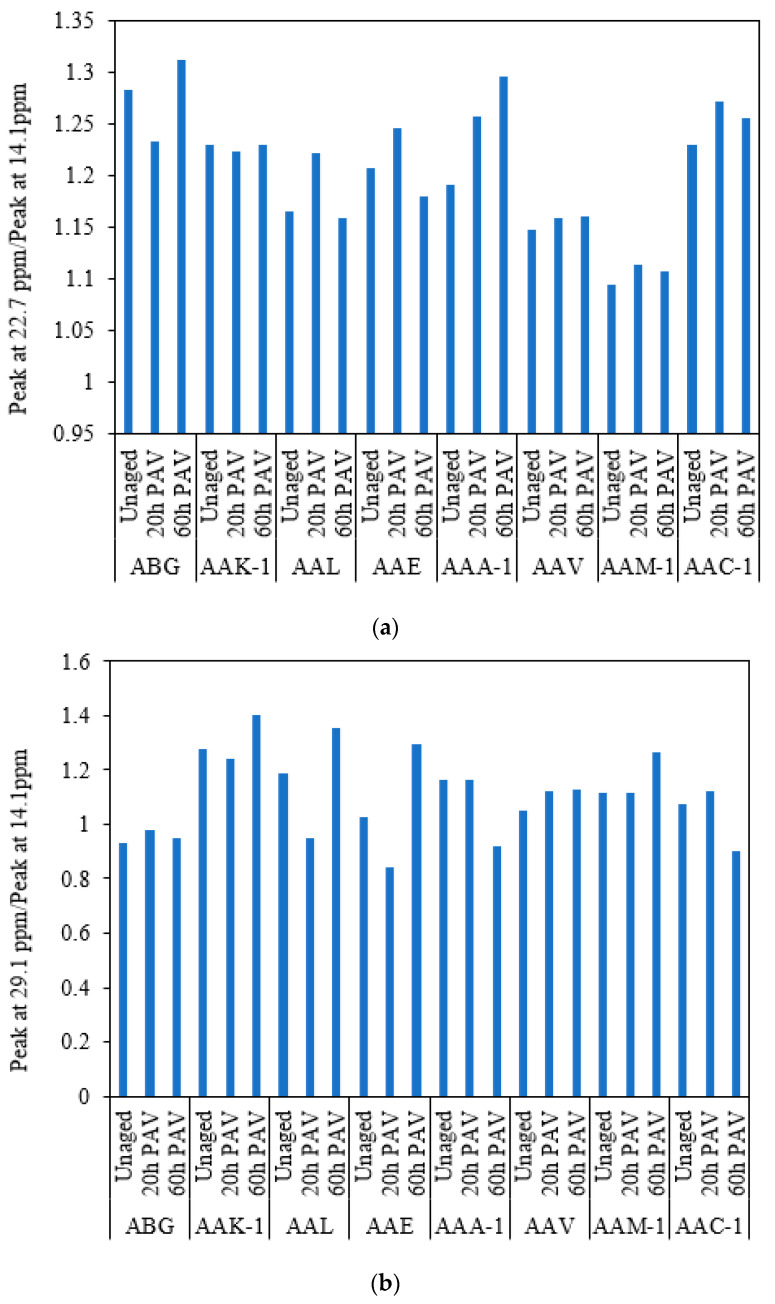
Percent carbons from ^13^C NMR for SHRP asphalt binders. (**a**) Peak@22.7 ppm/Peak@14.1 ppm (**b**) Peak@29.1 ppm/Peak@14.1 ppm (**c**) Peak@29.7 ppm/Peak@14.1 ppm (**d**) Peak@31.9 ppm/Peak@14.1 ppm.

**Figure 3 materials-15-08248-f003:**
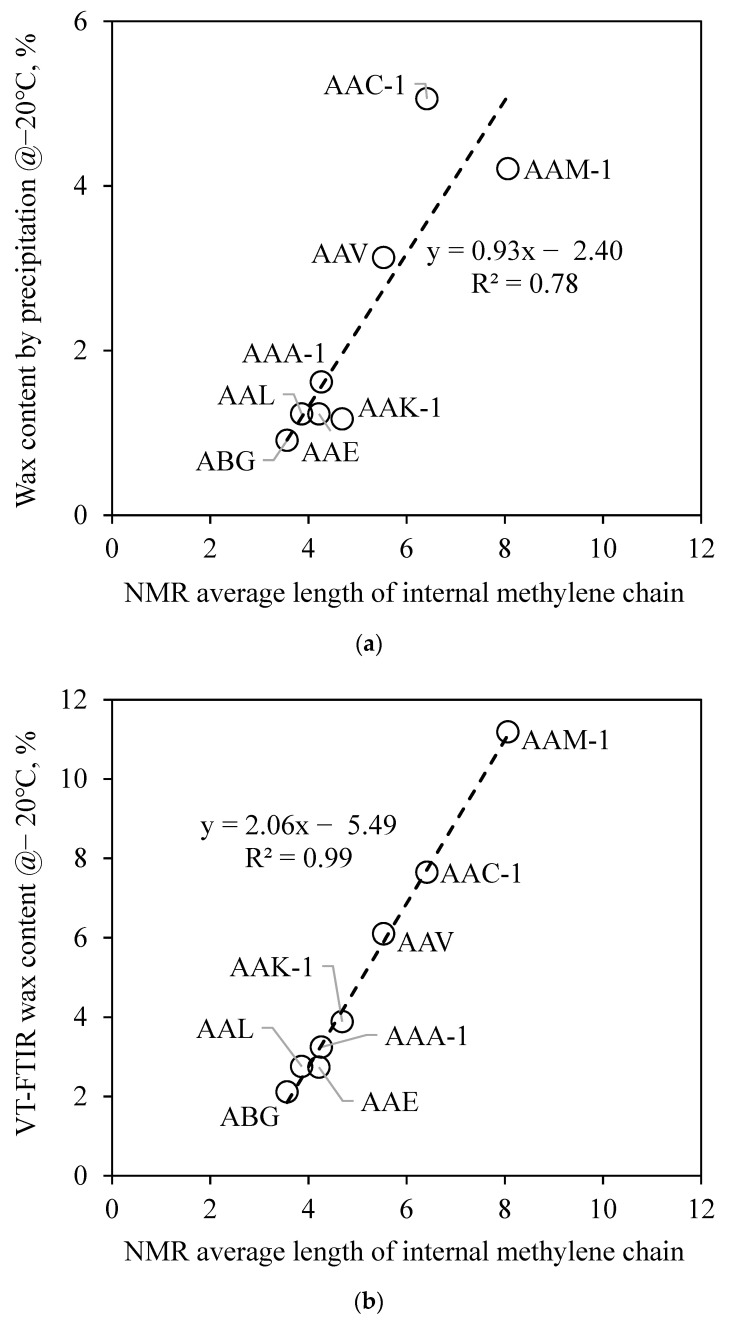
Comparison between the average length of straight internal methylene carbon of long paraffinic chain terminated by a methyl group and wax contents obtained using (**a**) precipitation and (**b**) VT-FTIR methods.

**Figure 4 materials-15-08248-f004:**
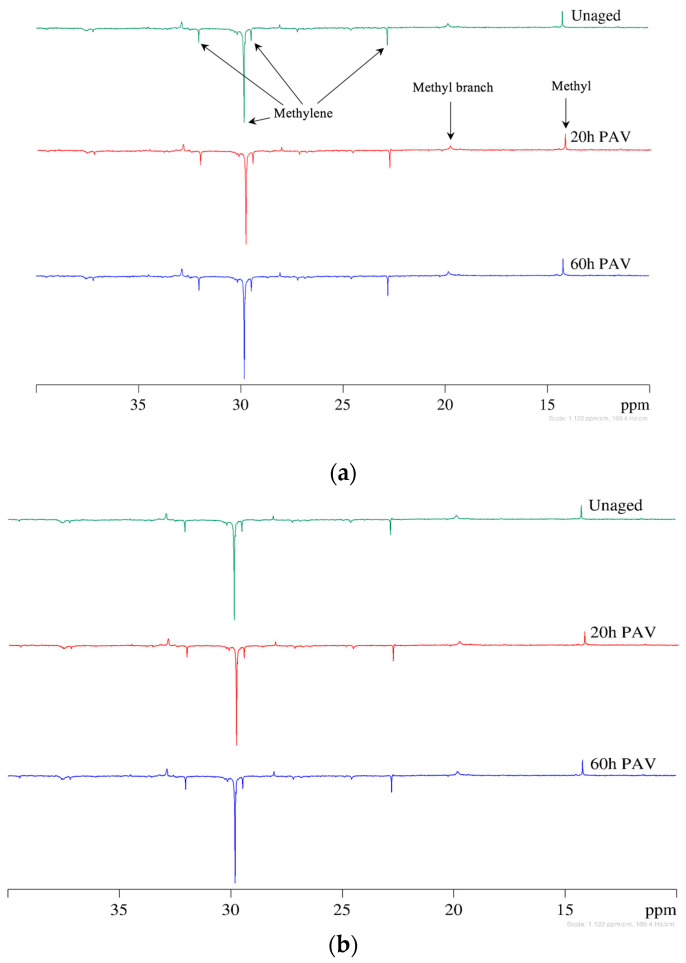
DEPT135 spectra of selected asphalt binders. (**a**) AAV. (**b**) AAC-1.

**Figure 5 materials-15-08248-f005:**
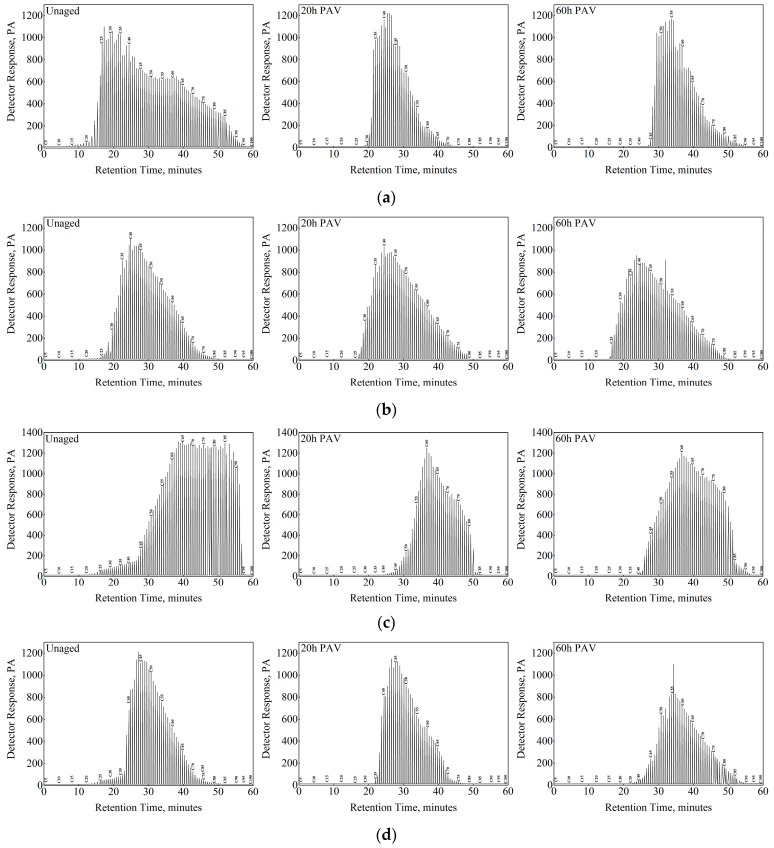
Gas chromatogram of asphalt binders. (**a**) ABG, (**b**) AAV, (**c**) AAM-1, and (**d**) AAC-1.

**Figure 6 materials-15-08248-f006:**
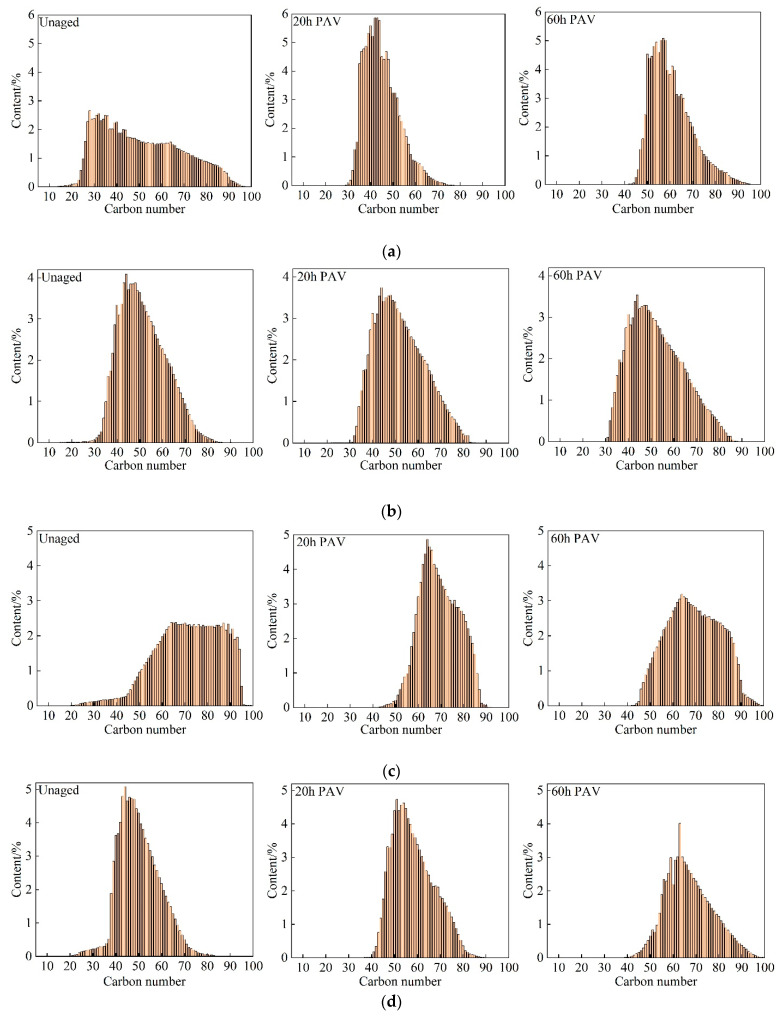
N-alkane distribution in asphalt binders. (**a**) ABG, (**b**) AAV, (**c**) AAM-1, and (**d**) AAC-1.

## Data Availability

The data supporting the figures in this paper, as well as other findings of this study, are available from the corresponding author upon reasonable request.

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
