# Peer review of "Assessment of Aging Impact on Wax Crystallization in Selected Asphalt Binders"

_materials, 2022, doi:10.3390/ma15228248_

Round 1

Reviewer 1 Report

The paper, entitled "Assessment of aging impact on wax crystallization in selected asphalt binders," addresses the problem of understanding changing trends in the crystalline components of asphalt binders. For this purpose, two advanced analytical methods were used, i.e. NMR and HTGC. The research conducted also included asphalts subjected to simulated PAV aging at two time intervals, i.e. 20h and 60h. Even in the title, the authors indicated that aging processes will be analyzed in the paper. Unfortunately, the topic undertaken was not fully expressed in the Introduction. The authors based their literature review of the available scientific literature on 23 papers (there are 26 total papers cited), where they focused mainly on asphalt waxes while completely ignoring the issue of aging asphalt binders. This omitted issue is an integral part of the work, so I don't know why it wasn't presented in adequate depth in the Introduction. Especially since the topic of aging of asphalt binders is now widely researched around the world, which is perfectly evident in the growing number of articles published on the subject. Therefore, before publishing the paper, it is necessary to supplement the Introduction with the topic of aging taking into account these works: https://doi.org/10.1016/j.conbuildmat.2018.10.169,  https://doi.org/10.1016/j.fuel.2022.124551, https://doi.org/10.1016/j.cscm.2020.e00390. The scope of the research conducted is wide. The authors analyzed 8 different asphalt binders with different PG and wax content. However, Section 2.1 lacks a description of the aging simulation or the standard according to which it was performed. Definitely such information should be added to this chapter. In addition, in my opinion, the 13C NMR analysis is incomplete. The 13C NMR spectra shown in Figure 1 cover the range from 0 to 40 ppm, which significantly narrows the structural analysis of the asphalt binders studied to aliphatic carbons only. Why did the authors not analyze a further range? In my opinion, these analyses should also be extended to other types of carbon atoms present in the structure of asphalt binders. The authors stated in line 168 that it is very difficult to capture the differences between the 13C NMR spectra obtained. I fully agree with this statement that merely observing the spectra does not lead to complete conclusions. Therefore, I believe that on the basis of the obtained spectra and the intensity of the observed bands, the authors should determine the percentage distribution of the different types of C atoms that build asphalt binders. These results would clearly and transparently indicate the structural differences of the asphalt binders studied. I also believe that the authors should correlate these results directly with PG and wax content. This would be very interesting and innovative for the reader. The authors in Figure 2 quantitatively analyze the 13C NMR spectra obtained, but I do not understand the approach of determining the ratio between peaks. Moreover, it is not clearly labeled what exactly the determined ratios describe. The authors should determine the percentage distribution relative to the sum of the intensities of all peaks visible on the spectrum. This approach is more correct in my opinion. I also do not understand why the peak at 14.1 ppm is used as a reference. I did not find such information in the text of the publication. In addition, Figures 5 and 6 are illegible. Their resolution and readability should be improved. 

In conclusion, the reviewed work is very interesting and definitely worthy of attention, but it still needs refinement and clarification of the inaccuracies described above. I recommend publication of the work after major revision. 

Reviewer 2 Report

Undoubtedly, the chemical structure of bitumen is of great importance, as research is currently being actively conducted, for example to find the most effective means of recovery when recycling asphalt concrete. Likewise, the chemical structure plays an important role, as partial substitutes for binders, for example biodegradable binders, are being sought. The interaction of the bitumen structure components with the regenerating/rejuvenation substance is a big challenge. Waxes make the bitumen brittle, so as it ages and increases in asphaltenes, the bitumen becomes brittle again. A synergy of two fragilities was formed. Elastic-viscous properties are recovered by adding maltene-containing substances. This article study waxes of natural origin, i.e. from crude oil. They appeared independently of us. However, waxes as an additive are a one of the recognized WMA additive, because faster than bitumen curing helps asphalt concrete to set faster and allows start traffic, while higher fluidity (low viscosity) of waxes than bitumen at high temeprature helps reduce production temperatures. But this is a case where the waxes are added in a controlled way. In order for this research to have greater applicability, as well as to relate aging and wax to real production/industry, it is recommend, for example, to refer to the DOI:10.3390/ma14143793. The conclusions should include more numerical values ​​so that it is clear what changes in percentages or absolute units were observed. Thus, future studies could envisage the evaluation of UV effects, the short-term RTFOT test, as well as the determination of rheological properties

Round 2

Reviewer 1 Report

The subject matter is within the scope of the journal.  The methodology is sufficiently well explained that someone else knowledgeable about the field could repeat the study. Each figure and table is necessary to the understanding of the conclusions. All elements of the manuscript relate logically to the study's statement of purpose. The manuscript have been improved and it can be accepted for publication.